# Paradoxical Role of AT-rich Interactive Domain 1A in Restraining Pancreatic Carcinogenesis

**DOI:** 10.3390/cancers12092695

**Published:** 2020-09-21

**Authors:** Sammy Ferri-Borgogno, Sugata Barui, Amberly M. McGee, Tamara Griffiths, Pankaj K. Singh, Cortt G. Piett, Bidyut Ghosh, Sanchari Bhattacharyya, Aatur Singhi, Kith Pradhan, Amit Verma, Zac Nagel, Anirban Maitra, Sonal Gupta

**Affiliations:** 1Department of Translational Molecular Pathology and Sheikh Ahmed Center for Pancreatic Cancer Research, University of Texas MD Anderson Cancer Center, Houston, TX 77030, USA; SFerri@mdanderson.org (S.F.-B.); sathiiitkgp@gmail.com (S.B.); Amberly.mcgee@gmail.com (A.M.M.); TGriffiths@mdanderson.org (T.G.); bghosh@mdanderson.org (B.G.); 2Department of Experimental Radiation Oncology, University of Texas MD Anderson Cancer Center, Houston, TX 77030, USA; Singh.pankaj@mayo.edu; 3Center for Radiation Sciences, Harvard University School of Public Health, Boston, MA 02115, USA; piett@hsph.harvard.edu (C.G.P.); znagel@hsph.harvard.edu (Z.N.); 4Department of Medicine, Albert Einstein College of Medicine, Bronx, NY 10461, USA; sanchari.bhattacharyya@einstein.yu.edu (S.B.); kith.pradhan@einsteinmed.org (K.P.); amit.verma@einsteinmed.org (A.V.); 5Department of Pathology, University of Pittsburgh Medical Center, Pittsburgh, PA 15232, USA; singhiad@upmc.edu

**Keywords:** SWI/SNF, pancreatic cancer, DNA repair, mouse model

## Abstract

**Simple Summary:**

Genes with deleterious mutations in tumors are widely accepted as tumor suppressors, since, loss of their normal expression often promotes tumor development. However, most tumors develop over a long period, with gradual accumulation of tumor-promoting events. Thus, to comprehend the role of individual genes in this evolving process of tumorigenesis, it is critical to investigate their role in both early precursors and established tumor cells. Despite recurrent mutations in *ARID1A* in genomes of human cancer, including pancreatic cancer, its role in tumorigenesis is not clear. We aim to understand the role of Arid1a in pancreatic cancer development and maintenance by investigating its role in both early pancreatic precursor cells and established pancreatic cancer cells. Besides novel understanding of context-dependent role of Arid1a in pancreatic cancer, this study will also enable development of therapeutic strategies for pancreatic cancers patients with *ARID1A* mutations, which is currently a critical unmet need in clinic.

**Abstract:**

*Background & Aims*: ARID1A is postulated to be a tumor suppressor gene owing to loss-of-function mutations in human pancreatic ductal adenocarcinomas (PDAC). However, its role in pancreatic pathogenesis is not clear despite recent studies using genetically engineered mouse (GEM) models. We aimed at further understanding of its direct functional role in PDAC, using a combination of GEM model and PDAC cell lines. *Methods*: Pancreas-specific mutant *Arid1a*-driven GEM model (*Ptf1a*-Cre; *Kras^G12D^*; *Arid1a*^f/f^ or “KAC”) was generated by crossing *Ptf1a*-Cre; *Kras^G12D^* (“KC”) mice with *Arid1a*^f/f^ mice and characterized histologically with timed necropsies. *Arid1a* was also deleted using CRISPR-Cas9 system in established human and murine PDAC cell lines to study the immediate effects of Arid1a loss in isogenic models. Cell lines with or without Arid1a expression were developed from respective autochthonous PDAC GEM models, compared functionally using various culture assays, and subjected to RNA-sequencing for comparative gene expression analysis. DNA damage repair was analyzed in cultured cells using immunofluorescence and COMET assay. *Results*: Retention of Arid1a is critical for early progression of mutant *Kras*-driven pre-malignant lesions into PDAC, as evident by lower Ki-67 and higher apoptosis staining in “KAC” as compared to “KC” mice. Enforced deletion of *Arid1a* in established PDAC cell lines caused suppression of cellular growth and migration, accompanied by compromised DNA damage repair. Despite early development of relatively indolent cystic precursor lesions called intraductal papillary mucinous neoplasms (IPMNs), a subset of “KAC” mice developed aggressive PDAC in later ages. PDAC cells obtained from older autochthonous “KAC” mice revealed various compensatory (“escaper”) mechanisms to overcome the growth suppressive effects of Arid1a loss. *Conclusions*: Arid1a is an essential survival gene whose loss impairs cellular growth, and thus, its expression is critical during early stages of pancreatic tumorigenesis in mouse models. In tumors that arise in the setting of ARID1A loss, a multitude of “escaper” mechanisms drive progression.

## 1. Introduction

The mammalian SWI/SNF chromatin remodeler subunit, AT-rich interactive domain 1 (encoded by the *ARID1A* gene), is believed to be involved in transcriptional regulation, DNA replication and DNA damage repair [1]. Recent integrated multi-platform sequencing analyses of pancreatic cancer ductal adenocarcinoma (PDAC) have revealed *ARID1A* mutations in ~6% of cases [2], besides predominant somatic mutations of *KRAS*, *TP53*, *SMAD4* and *CDKN2A*. These predicted loss-of-function alterations has led to the prevailing assumption that *ARID1A* behaves as a classic tumor suppressor gene (TSG), likely demonstrating genetic cooperation with mutant *KRAS* in pancreatic tumorigenesis. Based on this premise, recent genetically engineered mouse (GEM) models have been developed, with pancreas-specific *Arid1a* loss and mutant Kras expression [3,4,5,6]. Predominantly, these GEM models showed appearance of intraductal papillary mucinous neoplasms (IPMNs), a *bona fide* precursor lesion of PDAC, with only one study reporting progression to PDAC on the backdrop of IPMNs in 20% of these mice [3]. Other studies have also shown that the rate of *ARID1A* mutation or loss of expression in human IPMN samples is substantially higher than in human PDAC samples [3,5,7,8]. To functionally address this paradox, we revisited the role of *ARID1A* loss in multistep pancreatic carcinogenesis using GEM model and its direct cellular functions in cancer cells. Our functional data establishes that loss of ARID1A within pancreatic precursors restrains progression, unless circumvented by one of several escape mechanisms. Our findings reconcile the observed discrepancy in rates of Arid1a loss between human IPMNs and PDAC, and identify several potential targetable opportunities in tumors with *ARID1A* mutations.

## 2. Results

### 2.1. Loss of Arid1a Restrains Expansion and Progression of Ras-induced Pancreatic Precursor Lesions

To understand the functional role of Arid1a in development of pancreatic cancer, we generated *LSL-Kras^G12D^*; *Arid1a^fl/fl^*; *Ptf1a-Cre* (“KAC”) mice by crossing *Arid1a^fl/fl^*; *Ptf1a-Cre* (“AC”) mice with *Lox-stop-lox Kras^G12D^* mice [9]. The “AC” mice survived until euthanasia when 80-week-old but histology of pancreas showed characteristics similar to pancreatitis, starting at 8-weeks (Appendix A), progressing to additional fluid-filled macroscopic cysts, along with few focal low-grade PanINs (LG-PanINs) at 40-weeks (Appendix A). Immunohistochemistry (IHC) confirmed lack of Arid1a expression in epithelium of “AC” mice (Appendix A). Both “KAC” and littermate control “KC” mice (*LSL-Kras^G12D^*; *Ptf1a-Cre*) were necropsied either at first sign of distress or at periodic age intervals to perform histological assessment of pancreatic pathology. Numerous groups have described the natural history of disease in “KC” mice, which show histological pancreatic lesions, similar to human acinar-to-ductal metaplasia (ADM) and LG-PanINs around 16 weeks of age. These lesions progress to high-grade PanINs (HG-PanINs) with age and infrequently (<10% of cases) to frank PDAC after 1 year. In stark contrast with “KC”, necropsy of “KAC” mice revealed large fluid-filled cysts in “KAC” mice at 8 weeks, which increased in size and distribution with age (Figure 1A,B). Histological analysis showed extensive parenchymal replacement by mucinous cysts resembling low-grade branched duct gastric type IPMN (LG-IPMN) in humans, admixed with LG-PanINs adjacent to normal appearing parenchyma (Figure 1A).

With progressing age (20–32 weeks), the average area comprising of normal parenchyma gradually decreased in “KAC” pancreas (20.6% at 8–10 weeks, 7.7% at 16–20 weeks, 3.4% at 24–28 weeks), mostly replaced by LG-IPMNs, LG-PanINs and ADM lesions (Figure 1B). Lack of Arid1a expression in the epithelial precursor lesions arising in the “KAC” mice was confirmed by IHC (Appendix A). To assess the growth potential of precursor lesions in “KC” and “KAC” mice, we performed IHC for Ki-67 on age-matched pancreatic tissues. As expected in “KC” mice, an average of 40–50% of PanINs showed nuclear Ki-67 staining, in contrast with ~5% in IPMNs and PanINs of “KAC” mice, indicating lower proliferative potential of these cells (Figure 1C, Appendix A). This lower proliferative state was complemented by strong staining for cleaved caspase-3 in precursor lesions in “KAC” pancreata, indicating ongoing apoptosis in these cells (Figure 1D, *Early*; Appendix A, *Late*). This suggested that Arid1a is critical for growth of pancreatic premalignant lesions during early stages of disease development.

Despite restrained growth of precursor lesions, we witnessed a sudden shift in mortality of “KAC” mice with subsequent age (Figure 1E; 84% survival at 32-weeks versus 48% at 38-weeks). Histological assessment of pancreatic tissue revealed isolated foci of well-differentiated PDAC surrounded by confluent areas of LG-IPMNs, scattered ADMs and LG-PanINs in 4/11 mice (Figure 1F). With progressing age, survival of “KAC” mice reduced further to 36%, accompanied by growth of focal PDAC in 18/30 “KAC” mice, which were topographically distinct from LG-IPMN, LG-PanINs and rest of pancreatic tissue. Most of these PDAC were well-differentiated and metastasized to liver and lung (Figure 1G). Notably, in these older mice, PDAC were still accompanied by LG-IPMN and LG-PanINs, instead of high-grade precursor lesions, which was in stark contrast with stepwise progression of lesions with age from low- to high-grade PanINs in “KC” mice.

### 2.2. Loss of ARID1A Expression in Human IPMNs with Low-Grade Dysplasia

Due to dominance of LG-IPMNs of gastric subtype in Arid1a-null “KAC” mice, we assessed for loss of expression of ARID1A by IHC in a collection of 53 human IPMN sections representing various histological subtypes and dysplasia grades. Notably, loss of ARID1A expression was restricted only to low grade IPMNs of gastric subtype (Figure 2A,B, Appendix A) in 10/53 of all cases (~19%), while it was uniformly retained in high-grade IPMNs of gastric, intestinal and pancreato-biliary subtypes. Indeed, the high-grade IPMNs had an even stronger intensity of nuclear ARID1A expression than seen in adjacent normal ductal epithelium, suggesting an ongoing requirement for Arid1a function, in order to manifest as dysplastic progression. Even in IPMNs cases with mixed histological grades, the low-grade epithelium lacked ARID1A expression while the region with progression to high-grade dysplasia retained strong expression (Figure 2C). These findings from human IPMNs supported our GEM model data that loss of Arid1a might paradoxically constrain the progression of low grade IPMNs into higher-grade lesions. 

### 2.3. Arid1a Loss is Detrimental to the Growth of Established PDAC Lines

To assess the direct functional role of Arid1a without other compensatory molecular events appearing during autochthonous progression in the “KAC” GEM model, we employed CRISPR-Cas9 based approach to delete this gene in the Arid1a-expressing PDAC lines established from spontaneous PDAC arising in “KC” and “KPC” (*LSL-Kras^G12D/+^*; *LSL-Trp53^R172H/+^*; *Pdx-1-Cre*) mice. Genomic PCR sequencing and immunoblot analysis confirmed *Arid1a* deletion in puromycin-selected clones of CRISPR-ARID1A “KC” and “KPC” cells (Appendix A and Figure 3A). Although CRISPR-ARID1A cells did not show any changes in cellular morphology (Appendix A), they were both significantly growth retarded in monolayer cultures (Figure 3B). Similarly, both “KC” and “KPC” CRISPR-ARID1A cells showed greater than 50% decrease in number of anchorage-independent colonies (Figure 3C,D). Cell cycle analysis showed that *Arid1a* deletion caused significant G1 arrest in both “KC” and “KPC” cells (Figure 3E). This growth defect of CRISPR-ARID1A in both “KPC” and “KC” cells was also conspicuous in an orthotopic mouse model, where no macroscopic tumor was visible 7 weeks after implantation (Figure 3F, “KPC”). Histological analyses confirmed absence of any ARID1A-negative PDAC cells in the residual pancreatic tissue and the only proliferating PDAC cells were ones that escaped *Arid1a* deletion (Figure 3G). To extend our findings from mouse model to human cells, we screened a panel of patient-derived low passage PDAC cell lines for ARID1A expression (Figure 3H) and deleted *ARID1A* in high ARID1A-expressing Pa04 cells, using CRISPR-Cas9 sgRNAs targeting two distinct portions of human *ARID1A* exon 1 (Figure 3I). Similar to mouse, CRISPR-ARID1A Pa04 cells showed growth retardation in both 2D and 3D growth assays compared to vector control (Figure 3J,K; Appendix A).

Despite our attempts to establish IPMN cell lines from “KAC” mice for further characterization, these cells failed to grow stably in culture, consistent with low proliferative potential of these cells in vivo. However, we were able to establish PDAC cell lines from an autochthonous primary tumor (““KAC”-P”) and the matched liver metastasis (““KAC”-L”) from a 12-month old “KAC” mouse with invasive cancer. For subsequent experiments, we also used the aforementioned parental “KC” and “KPC” lines as orthogonal controls for “conventional” PDAC. When comparing monolayer growth characteristics, we found “KC” and “KPC” cells grew in clustered colonies with tight cell–cell adhesion, autochthonous “KAC” lines showed elongated, spindle-shaped morphologies, characteristic of mesenchymal cells (Figure 4A). While these cells showed increased expression of epithelial–mesenchymal transition (EMT)-associated genes (Appendix A), the CRISPR-ARID1A derivatives of both “KPC” and “KC” cells showed reduced expression of EMT-associated genes (Appendix A) and reduced migration, compared to the control cells (Appendix A).

Similarly, in contrast with isogenic cell lines, there was no difference in growth of the autochthonous “KAC” cells with “KC” or “KPC” cells, in either monolayer or 3D cultures in soft-agar (Figure 4B,C), cell cycle progression (Appendix A) or in vivo growth in orthotopic mouse model, mirroring the well-differentiated grade of parental tumors (Appendix A). Thus, while Arid1a expression is critical for cellular growth, this requirement is bypassed in autochthonous PDACs arising in mice.

### 2.4. Identification of “Escaper” Mechanisms in Autochthonous Arid1a-null PDAC Cells

To elucidate potential “escaper” mechanisms that might allow autochthonous *Arid1a*-null “KAC” cells to circumvent the growth constraining effects of Arid1a loss, and progress to PDAC, we subjected the RNA from “KAC-P” and “KC” cells to global RNA sequencing (RNA-Seq). Among all differentially expressed transcripts (≥2-fold change, *p* < 0.05), Claudin 18 (Cldn18) was one of the most highly-expressed in “KAC” cells (Figure 4D, Appendix A), which was also confirmed at the protein level (Figure 4E). Interestingly, CLDN18, highly specific tight junction protein of the gastric mucosa, has been reported to be expressed in larger percentage of precursor lesions and indolent neoplasms, like PanINs and IPMNs, than in PDAC [10]. Indeed, we found robust CLDN18 expression in IPMN, PanIN and PDAC lesions from “KAC” mice (Figure 4F), and *ARID1A*-negative human PDAC tissues (Figure 4G); stressing the inverse correlation between the expressions of these proteins. ATAC-Seq revealed readily accessible chromatin surrounding the *Cldn18* regulatory locus in “KAC-P” *versus* “KC” cells suggesting Cldn18 transcription is directly altered via changes in chromatin accessibility upon *Arid1a* loss (Figure 4H). While we do not postulate CDN18 is an oncogenic “escaper” mechanism in “KAC” cells (especially as its upregulation is also seen in precursor lesions), our data identifies a facile membrane-based therapeutic target in human tumors with *ARID1A* mutations. We performed gene set enrichment analysis (GSEA) of differentially expressed transcripts for known tumor-promoting networks that might serve as putative “escaper” mechanism(s) in “KAC-P” cells. GSEA identified enrichment of signatures positively associated with EMT (validated in Appendix A), a positive association with Myc and E2F activity and negative association with p53 and Ras signaling in “KAC”-P” cells (Figure 5A,B; Appendix A). We confirmed loss of Arid1a and p53 protein expression, and diminished expression of Cdkn1A/p21 (canonical p53 target), phospho-p44/42 MAPK (ERK1/2) and total ERK1/2 (downstream targets of oncogenic Kras signaling) in “KAC” lines, compared to “KC” control (Figure 5C, Appendix A). Further, consistent with the Myc activation signature in “KAC-P” cells, we noted a substantial increase in levels of *TRP63* RNA and protein expression of ΔNp63α and γ isoforms in “KAC” cells compared to “KC” and “KPC” cells (Figure 5C, Appendix A), which has been recently shown to positively regulate MYC function [11]. PDAC sections from the “KAC” mice also showed loss of p53 (Figure 5D) and strong MYC expression (Figure 5E), suggesting these as putative “escaper” mechanisms in the face of Arid1a loss. Parsing the RNA-Seq data, we identified pluripotency-associated transcription factors—Sox2 and Nanog—as significantly overexpressed in “KAC-P” cells (Appendix A) and confirmed high Sox2 expression and open promoter in “KAC” cells compared to control “KC” (Figure 5F,G). Both transcription factors are aberrantly expressed in multiple cancers, including PDAC [12,13,14] and believed to mark cancer stem cells and promote EMT [15], suggesting that transcription factors implicated in stem cell identity might also play a role in development of PDAC in “KAC” mice.

### 2.5. Impaired DNA Damage Repair as a Potential Mechanism Restraining Neoplastic Progression in Arid1a Null Cells

Prior studies have shown that SWI/SNF complexes often localize to sites of DSBs and facilitate chromatin decondensation following serine 139 phosphorylation of histone H2AX (P-H2AX) via ATM/ATR [16]. Surprisingly, RNA-Seq data showed enrichment of gene signatures associated with enhanced DNA repair in “KAC-P” compared to “KC” cells (Appendix A). Underscoring this paradox, we found “KAC-P” cells to be relatively resistant to Cisplatin, compared to “KC” and “KPC” cells (Appendix A). Similarly, contrary to other reports [17,18], “KAC” and ARID1a-deleted isogenic “KC” and “KPC” cell lines were also resistant to ATR inhibitor, either alone or in context of DNA damage (Appendix A) or to various PARP inhibitors. This led us to postulate whether in the compendium of potential “escaper” mechanisms that lead to emergence of cancers in “KAC” mice, overcoming an inherent DNA repair defect could also be one, thereby providing survival advantage to a subset of *Arid1a-null* cells, and a permissive milieu for progression to PDAC. To test this hypothesis, we assessed DNA repair competency upon immediate Arid1a deletion, mediated by CRISPR-Cas9 in “KPC” and “KC” cells. Doxorubicin (Doxo) is a well-known chemotherapeutic agent known to induce DSBs and early activation of ATM leading to pH2AX and p53 activation [19]. Thus, pH2AX is an early and sensitive marker of DSB induction after Doxo treatment, when measured as foci by immunofluorescence [20], along with foci formation by DNA damage-responsive protein 53BP1 [21]. When measured in Doxo-treated isogenic “KPC” cells with and without ARID1A expression, we found significantly reduced levels of pH2AX and 53BP1 foci in CRISPR-ARID1A compared to CRISPR-EMPTY cells, even at baseline (Figure 6A and Appendix A). After 2 h of Doxo treatment, while control cells showed increase and subsequent drop to baseline state after 24 h in pH2AX and 53BP1-positive foci; the levels of foci in CRISPR-ARID1A cells were always significantly lower than control “KPC” cells, suggestive of an impaired DNA damage response (DDR). However, when assessed in the setting of wild type *TRP53* using isogenic “KC” cells, we did not find this decrease in the levels of either pH2AX or 53BP1 foci upon ARID1A deletion (Figure 6B, Appendix A). For further evaluation of DNA damage and repair, we performed an orthogonal and highly sensitive comet assay, wherein quantification of the comet tail intensity relative to the head (% of DNA in tail) reflects the number of DNA breaks. Control cells from both “KPC” and “KC” cell lines (CRISPR-EMPTY) showed an initial increase of comets at 2h followed by reduction at 24 h post-Doxo treatment, suggesting repair of damaged DNA (Figure 6C–F). In contrast, CRISPR-ARID1A cells from both lines showed higher amount of damaged DNA at 24 h (Figure 6C–F). This suggests that ARID1A is critical for DNA damage repair, irrespective of active p53 signaling, but redundant for initial P-H2AX or 53BP1 foci formation in presence of functional p53. Validating our premise of an “escaper” phenomenon in autochthonous “KAC” lines, these cells while showing diminished levels of pH2AX or 53BP1 foci (Appendix A), reflecting their origin in the setting of *Arid1a* deficiency, demonstrated competency at post-Doxo DNA repair comparable to both “KC” and “KPC” lines (Appendix A).

### 2.6. Loss of Arid1a Causes Impaired Mismatch Repair (MMR) in PDAC Cells

Recently, ARID1A was suggested to be important for DNA mismatch repair (MMR) due to its interaction with the MMR protein, MSH2 in ovarian and colon cancer cell lines [22]. Thus, we examined the direct role of ARID1A in MMR using the isogenic “KC” and “KPC” cell lines. ARID1A loss in both “KC” and “KPC” lines showed compensatory increase in expression of MSH6, PMS2 and MLH1 proteins, suggestive of impaired MMR function and an ongoing requirement for the MMR machinery (Figure 6G). Notably, overexpression of PMS2 has been shown to disrupt mammalian MMR function causing genetic instability [23]. To confirm impaired MMR function, we utilized a quantitative functional MMR reporter assay [24] and found reduced MMR capacity in the isogenic “KPC” line upon ARID1A deletion (Figure 6H). In contrast to the discordance between autochthonous and isogenic lines observed with DDR, both “KAC” cell lines also showed stronger expression of MSH6 and PMS2 proteins (Figure 6I), and functional impairment of MMR on the reporter assay compared to “KC” and “KPC” cells (Figure 6J). Thus, in contrast to the defective DNA repair that appears to be at least partially responsible for restraining neoplastic progression within the pancreatic epithelium, impaired MMR in the setting of ARID1A loss appears to be a tumor neutral phenomenon that persists in the autochthonous lines.

### 2.7. Synthetic Lethal Targeting of Arid1a Loss in PDAC

The concept of synthetic lethality has been widely exploited for cancer therapy since most cancers have loss-of-function mutations that are not readily targetable [25]. ARID1B, a structurally related but mutually exclusive homolog of ARID1A in the SWI/SNF chromatin-remodeling complex, is a potential synthetic lethal vulnerability in ARID1A-mutant human cancers [26]. Indeed loss of ARID1B in ARID1A-deficient cells destabilizes the SWI/SNF complex and impairs proliferation in gastric and ovarian cancers [27]. To explore this vulnerability in PDAC using our GEM model, we first confirmed the expression of ARID1B in ARID1A-null cell lines. Remarkably, both “KAC” lines and isogenic lines with ARID1A deletion (CRISPR-ARID1A) showed increased expression of ARID1B, at both mRNA (Figure 7A,B) and protein level (Figure 7C,D), suggestive of a compensatory increase upon ARID1A loss. After confirming the successful knock down of Arid1b expression using short hairpin (pLK0-shARID1B) in both autochthonous and isogenic cells (Figure 7C,D), we found that only the “KAC” cell lines showed ≥ 80% decrease in cell proliferation, compared to “KC” or “KPC” control cells upon Arid1b knockdown (Figure 7E). Similarly, only ARID1A-null “KC” and “KPC” isogenic cell lines (CRISPR-ARID1A) showed reduction in cell proliferation compared to CRISPR-EMPTY control upon co-extinction of ARID1B (Figure 7F). The effect of Arid1b knockdown was also remarkable in reducing anchorage-independent growth of “KAC” cell lines in 3D cultures (~70% reduction in colony count; Figure 7G and Appendix A). This indicates that ARID1B is a potential therapeutic target in ARID1A-deficient PDAC tumors, although currently there are no specific ARID1B inhibitors available for clinical trial. To overcome this concern, we explored other potential synthetic vulnerabilities of ARID1A-loss in our “KAC” model using commercially available small molecule inhibitors. Since MYC was one of the key “escaper” pathways upregulated in our “KAC” model, we tested the therapeutic vulnerability of “KAC” cells using 10058-F4, a specific small molecule inhibitor of MYC that prevents transactivation of MYC target gene expression [28]. In both monolayer and 3D spheroid cultures, “KAC” cell lines were significantly more vulnerable to 10058-F4 than control “KC” cells (Figure 7H, Appendix A). This “onco-dependence” suggests MYC activation to be, at least in part, critical for survival of “escaper” ARID1A-null cells. Recently, *Arid1a* mutation was reported as a biomarker for sensitivity of platinum-resistant urothelial carcinoma cells to Panobinostat-mediated HDAC targeting [29]. Interestingly, GSEA of gene expression data in these cells showed enrichment for MYC, E2F targets, and DNA repair pathways, similar to our observed GSEA data in “KAC” cells. This provided us with a strong rationale to test synthetic lethality of panobinostat in our “KAC” PDAC model. Remarkably, compared to control “KC/”KPC” cell lines, “KAC” cells showed significantly better sensitivity to panobinostat in both 2D monolayer cultures (Figure 7I) and 3D cultures (Appendix A). Since panobinostat is already a clinic-ready drug, this presents another potential opportunity for targeting *Arid1a*-null PDAC tumors.

HOMER analysis of our ATAC-Seq data showed enrichment in binding sites for PU.1 transcription factor in open chromatin regions of “KAC” compared to “KC” cells (Appendix A), suggesting higher functional activity of PU.1. Indeed, RNA-Seq and RT-PCR in the same population showed significant increase in levels of Csf1 (Appendix A), which is a bona-fide target of PU.1. Utilizing a first-in-class small-molecule PU.1 inhibitor that specifically and allosterically interfere with PU.1-chromatin binding [30], we found “KAC” cells were significantly more sensitive to growth inhibition, compared to control “KC” and “KPC” cells (Figure 7J), presenting PU.1 as another synthetic lethal target in ARID1A-null PDAC cells.

In summary, we identified a novel context-dependent role of Arid1a in PDAC where immediate loss of Arid1a function in cells is growth restrictive, at least partially due to impaired DNA repair, but potentially creates opportunities for various compensatory oncogenic mechanisms to drive the disease progression. We also identified therapeutic vulnerabilities of Arid1a-mutant PDAC cells that can be readily tested in clinical studies in imminent future.

## 3. Discussion

Deleterious mutations of *ARID1A* reported across tumor types, led to the prevailing assumption that it behaves as a prototypal tumor suppressor. However, DepMap analysis (https://depmap.org/portal/) of ARID1A using both CRISPR and RNAi screens in various cancer and pancreatic cancer cell lines showed dependency scores (CERES) < 0, which indicates it is an essential gene for survival, comparable to the median of all pan-essential genes. Nonetheless, emerging evidence from both functional and correlative data challenges the role of *Arid1a* as a relatively straightforward TSG. For example, Zhu and colleagues demonstrated in murine models of hepatocellular carcinomas (HCCs) that sustained Arid1a expression was pro-tumorigenic, while loss of Arid1a was deleterious, during primary tumor formation [31]. In fact, this and other studies [32] have shown that the vast majority of primary human HCCs (85–90%) retain Arid1a expression, at levels greater than the background liver, reiterating a need for sustained, and potentially enhanced Arid1a functional requirement, in early hepatocarcinogenesis. Similarly, concomitant bi-allelic deletion of Arid1a in the Apc*^min^* mice significantly inhibited intestinal neoplasia [33]. In addition, CRISPR-mediated deletion of *ARID1A* in human colorectal cancer cells with *KRAS* mutations significantly reduced proliferation, accompanied by attenuation of MEK/ERK dependent transcriptional signaling [34]. *ARID1A* mutations also correlated with better survival in uterine corpus endometrial carcinoma [35]. Similarly, in the context of pancreatic neoplasia, single cell analysis of human IPMNs showed 40% with subclonal *ARID1A* mutations, all in the low-grade gastric type IPMNs [7], in contrast with 6% somatic *ARID1A* alterations in PDAC reported in TCGA analysis [2]. This supports our present finding of complete loss of ARID1A protein expression in ~20% of patient IPMN samples, all occurring in low-grade gastric type IPMNs. In the COSMIC database as well, well-differentiated indolent pancreatic neuroendocrine tumors, with a proliferation rate < 3% Ki-67 [36], carry a ~20% ARID1A mutation rate, much higher than 5.35% aggressive PDAC. Notably, these findings are in sharp contrast to well-established TSGs, like *TRP53* and *CDKN2A*, which demonstrate a progressive increase in rate of alterations from low-grade to high-grade precursors to invasive adenocarcinoma [37]. These lines of evidence suggest that loss of Arid1a might not demonstrate outright genetic cooperation with oncogenic Ras in the pancreatic epithelium and thus warrant a careful reappraisal of the TSG role for *ARID1A* during multistep neoplastic progression ascribed in recently published GEM models [3,4,5,6].

Indeed, our genetically-engineered mice of conditional *Arid1a* loss with co-expression of mutant *Kras*^G12D^ allele (“KAC”), developed LG-IPMNs and LG-PanINs ubiquitously, with the former resembling gastric type IPMNs in patients, as has been previously reported [5,6]. In contrast to “KC” mice, there was no stepwise progression of the LG precursors to HG precursor lesions, along with low proliferative index and higher frequency of apoptotic nuclei in the “KAC” pancreas. Surprisingly, another study also documented this low proliferation rate induced by *Arid1a* loss in pancreatic precursor lesions [3], albeit characterizing their IPMNs to be of pancreatobiliary- or oncocytic subtype, in contrast with gastric subtype reported by us and others. In attempt to understand how some precursor cells break this low-grade dormancy to progress into aggressive PDAC, we established cell lines from primary and metastatic PDAC lesion from the aged “KAC” mice. Transcriptome analysis on these paired lines demonstrated several signaling nodes that were aberrant compared to “KC” cells, including downregulation of p53, upregulation of MYC, EMT- and pluripotency-associated transcription factors. Interestingly, MYC transcription has been reported to be repressed by p53 and the loss of p53 synergistically enhancing the Myc–induced tumorigenesis [38]. The appearance of these highly ranked aberrant signaling nodes was plausible, since other autochthonous models have also reported upregulation of EMT-associated genes [6] and MYC activation within the resulting *Arid1a*-null cancers [5]. Low ARID1A expression also significantly correlated with low Ki-67 labeling index and negative p53 expression in breast cancer patients [39]. Interestingly, Ras signaling was downregulated in the “KAC” lines, which was confirmed by assessment of MAPK activity. In this context, while MYC is considered a downstream effector of oncogenic Ras in PDAC, mediating its pleiotropic effects on tumor cell growth and metabolism [40,41], MYC can also substitute as a pivotal driver in the setting of “Ras independence”, and the resulting tumors tend to be highly aggressive and chemoresistant [42,43].

The isogenic CRISPR-ARID1A clones of “KC” and “KPC” cells were strikingly different in their behavior from the autochthonous “KAC” PDAC lines, which can be postulated to arise through an “escape” phenomenon from *Arid1a* deletion-induced growth constraint in vivo, likely under the selection pressure of oncogenic Ras and other secondary events within the pancreatic epithelium. Further, the compendium of tumor-promoting pathways in the autochthonous PDAC models (such as Myc upregulation and perturbation of p53 function) identified by us, and others [3,5,6], are likely to be the molecular adaptations underlying this “escape” phenomenon. It is worth noting that in at least one prior study [6], the “KAC” genotype rarely progressed to invasive cancers, with mice mostly developing IPMN precursors, unless additionally crossed to a mutant *Trp53* background (providing one prototypal “escape” mechanism). Similarly, our recently published CRISPR-based mouse model of PDAC [44] showed *Arid1a* loss concurrent with oncogenic *Kras* mutation in adult acinar tissue only caused LG-PanINs, while emergence of well-differentiated PDAC in the same period required deletion of *TRP53*, irrespective of Arid1a loss. The distinction between the two scenarios—whether Arid1a loss cooperates with oncogenic Ras to induce PDAC formation (as proposed [3,4,5,6]), or invasive cancers arise via an “escape” phenomenon in the setting of growth constrained precursors–goes beyond semantics, given the occurrence of *ARID1A* mutations across a multitude of epithelial pre-cancers [45].

What mechanism underlies the inability of *Arid1a*-deleted precursor lesions to robustly proliferate in vivo? GSEA data on the “KAC” PDAC lines identified DNA repair as one of the highly ranked pathways, which was paradoxical, since numerous prior studies in preclinical models have suggested ARID1A protein to be integral to repair of DNA DSBs [16,18,46,47]. Consequently, cell lines with *ARID1A* mutations showed increased sensitivity to cisplatin, radiation and PARP inhibitors [18,47]. Surprisingly, our “KAC” cells were both relatively resistant to cisplatin and PARP inhibitors (compared to the “KC” and “KPC” lines), and by the Comet assay, their DNA repair proficiency was comparable to these *Arid1a* wild type PDAC lines. This led us to postulate that acquisition of DNA repair capacity might be a crucial “escape” mechanism co-opted by “KAC” cells, and *vice versa*, impaired DNA repair a feature of *Arid1a* ablated precursors. Indeed, in response to doxorubicin-induced DNA damage, isogenic derivatives of “KC” and “KPC” cells showed compromised DNA repair on the Comet assay, compared to the respective parental controls. Extrapolating from these findings, we suggest that early PDAC precursors (LG-IPMNs and LG-PanINs) with Arid1a loss are vulnerable to genetic stress within the pancreatic epithelium created by mutant Ras generated reactive oxygen species [48,49]. During the natural history of these mice, *Arid1a*-null precursor lesions that are unable to adapt, either “stall” or undergo apoptosis, while clones that can re-functionalize their repair machinery through secondary adaptations (e.g., loss of p53 expression) survive and subsequently progress to PDAC.

Finally, an area of considerable translational potential is the opportunity to develop targeted therapies against PDAC harboring *ARID1A* mutations. In light of our observations, it is imperative that putative targets be distinguished into those that *persist* in the established PDAC, *versus* those present in precursor lesions, but *circumvented* in established PDAC. For example, we posit that “escape” mechanisms are the basis for our observation that synthetic lethal effects described in other *ARID1A* mutant solid cancer models, such as susceptibility to PARP inhibition and cisplatin [18,46,47], are absent in the autochthonous PDAC lines emerging in the setting of “escape”. On the contrary, a striking “synthetic essentiality” [25] identified in the *Arid1a* mutant PDAC cells is the requirement of sustained ARID1B expression for survival. Increased expression of Arid1b in *Arid1a*-null cells irrespective of an “escape” setting (i.e., in both “KAC” cells and the CRISPR-isogenic cells), points towards a compensatory effect directly related to *Arid1a* deletion. GeneHancer analysis [50] of promoter/enhancer region of *Arid1b* showed multiple binding sites for transcription factors such as MYC (11 sites), RUNX3 (25 sites), SOX6 (14 sites), FOXA1 (nine sites), and ZNF213 (seven sites). Interestingly, expression of these transcription factors was also upregulated (Appendix A, RNA-Seq) and their binding sites associated with open chromatin, as identified by motif analysis of ATAC-seq data) in “KAC” cells (Appendix A). We also identified promoter/enhancer regions of both *ARID1B* and *CLDN18* replete with binding sites for the Pu.1 (SPI1) transcription factor (e.g., the *ARID1B* promoter/enhancer had as many as 18 binding sites). Pu.1 could be one of the pivotal transcription factors driving the compendium of transcriptional alterations directly related to *ARID1A* loss in the pancreatic epithelium. Indeed, our proof of concept data with a first-in-class Pu.1 small molecule inhibitor demonstrates robust growth inhibition in “KAC” cells, and pending future validation studies, this might represent a facile pharmacological strategy for targeting *ARID1A* mutant PDAC. Additionally, Panobinostat has been shown to cause growth arrest and apoptosis in cells from many cancer types including leukemia, by decreasing MYC expression and increasing expression of TRP53, CDKN1A (p21) and DNA repair genes [51]. In ovarian cancer model, ARID1A loss inactivated the pro-apoptotic function of TRP53 by upregulating HDAC6 [52]. Thus, it was not surprising that Panobinostat was particularly effective against Arid1a-null “KAC” cells with elevated Myc signaling and decreased TRP53 and CDKN1A expression.

In conclusion, using a repertoire of GEM models, autochthonous and isogenic cell line models, we provide compelling evidence that *ARID1A* loss induces a paradoxical growth constraint within low-grade cystic precursor lesions harboring mutant Ras; a finding we believe underlies the prolonged, indolent natural history of most gastric type human IPMNs. Eventually, through loss of p53, or upregulation of oncogenic networks like Myc and pluripotency factors, a subset of Arid1a-depleted precursor cells progress to frank adenocarcinomas. Our data reassesses the utility of therapeutic vulnerabilities previously described in other *ARID1A* mutant cancer, in the setting of PDAC [18,46,47], while describing novel opportunities for targeting this class of cancers in the clinic.

## 4. Materials and Methods

Additional methods are described in the Appendix A.

### 4.1. Cell Culture

Murine “KPC” cell line was derived from a spontaneous tumor arising in a female *LSL-Kras^G12D/+^*; *LSL-Trp53^R172H/+^*; *Pdx-1-Cre* (“KPC”) mouse; “KC” and “KAC” cell lines were isolated from spontaneous tumors arising in *LSL-Kras^G12D/+^*; *Pdx-1-Cre* and *LSL-Kras^G12D/+^*; *ARID1a^fl/fl^*; *Ptf1a-Cre* mice, respectively and their epithelial origin confirmed by genomic PCR for Cre-mediated Kras recombination [3,4,5,6]. Patient-derived low passage cell line Pa04 was cultured as described before [53]. Cells were maintained at 37 °C in humidified 5% CO_2_ incubator and cultured in DMEM (Cat#D6429, Sigma-Aldrich, St. Louis. MO, USA) supplemented with 2 mM L-glutamine (Cat#G7513, Sigma-Aldrich), 10% FBS (Cat#F2442, Sigma-Aldrich), and 100 µg/mL Pen-Strep (Cat# 30002CI, Corning, Corning, NY, USA). All cell lines were tested routinely for mycoplasma contamination.

### 4.2. ARID1A Deletion by CRISPR/Cas9

sgRNA cassette was generated using the CRISPR design tool (http://crispr.mit.edu). The sequence of various sgRNAs were Mouse-Ex.2a sgRNA (GGTCCCTGTTGTTGCGAGTA), Mouse-Ex3a sgRNA (GCCCTGCTGGCCATACGCAC), Human-Ex1a sgRNA (GATCCCCGCTGTCTC GTCCG), Human-Ex1b sgRNA (TTGTTGGGCCCCTCCCGAGG). The sgRNAs were cloned into the pX459 (pSpCas9 (BB)-2A-Puro) plasmid vector (Cat#62988, Addgene, Watertown, MA, USA). Mouse or human cell lines were transfected with the abovementioned plasmids using Lipofectamine 3000, and positive cells selected in the presence of 1.5 µg/mL puromycin.

### 4.3. Determination of Cell Growth and Morphology

Cellular morphology and proliferation were assessed using Incucyte live cell imager (Sartorius, Bohemia, NY, USA) and images analysed with the IncuCyte HD software. Proliferation was measured through quantitative kinetic processing metrics derived from time-lapse image acquisition and presented as percentage of culture confluence over time. For experiments where cells underwent change in morphology due to effect of a drug, cells stably expressing the inert RFP in the nuclei (transfected with IncuCyte NucLight Red Lentivirus Reagent; Cat#4476, Sartorius, Goettingen, Germany) were used to perform quantitative kinetic metrics and evaluate proliferation, expressed as count/well.

### 4.4. 3D Cultures for Colony Formation

Anchorage-independent growth assay in soft-agar was performed as described before [53]. Briefly, cells were seeded at a density of 1000 cells/well in 12-well plates with drug treatments on the next day. After two weeks, plates were fixed and stained with 0.005% Crystal violet solution (Sigma-Aldrich, Cat#C6158), imaged with ChemiDoc scanner (BioRad, Hercules, CA, USA) and colonies counted using ImageJ software. For spheroid cultures, log-phase cultures were seeded @ 1000 cells/well in ultra-low attachment, round-bottom 96 well plates (cat #7007, Corning Costar, Corning, NY, USA) and allowed to grow in 37 °C humidified growth incubator for 7–10 days. For growth inhibition assays, treatment with chemical inhibitors or corresponding vehicle control was done 24 hr after seeding and spheroid growth imaged using spheroid imaging protocol of the Gen5 Image software on Cytation 3 Cell Imaging Multi-mode Reader (Agilent Technologies, Santa Clara, CA, USA) using 4× magnification, N.A. 0.1 and ACH as level of correction of the objective lens.

### 4.5. Migration Assay

In 96-well Image Lock plates (Cat#4379, Sartorius, Goettingen, Germany) of confluent cell culture, a central scratch-wound per well was made using the 96-pin WoundMaker (Cat#4493, Essence BioScience). Cells were grown for a further 24 h and the recovery of the scratch-wound was analysed by taking images at 1 h intervals with the Incucyte Live-Cell Imaging System (Essence BioScience). The images were analysed with the IncuCyte HD software and the results presented in the form of relative wound densities and standard deviations for each time point. The relative wound density (%) represents the cell density in the wound area expressed relative to that outside the wound area as a function of time.

### 4.6. Immunofluorescent (IF) Staining for Foci Formation

Twenty-four hours after seeding cells on IBIDI µ-Slides Collagen IV coated (Cat#80822, IBIDI, Fitchburg, WI, USA), they were treated with 0.1 µM of Doxorubicin (Doxo) (Cat#D1515, Sigma-Aldrich) for 30 min. After media change, cells were grown for 0, 2, 4, 8 and 24 h and then washed with DPBS (Cat#D8537, Sigma-Aldrich) and fixed with 4% formalin (Cat#HT5011, Sigma-Aldrich) for 10 min at RT. After washes, cells were Permeabilized with DPBS containing 0.5% Triton-X100 (Cat#X100, Sigma-Aldrich) for 10 min at RT. Blocking was then performed with DBPS containing 3% BSA (Cat#311695600, Roche, Basel, Switzerland) and 1% Chicken serum (Cat#C5405, Sigma-Aldrich) for 1 h at RT and P-H2AX (1:800) and 53BP1 (1:1000) primary antibodies were incubated over night at 4C and secondary antibodies for 1 h at room temperature. After DAPI staining, slides were mounted with mounting media (Cat#S3023, Agilent Technologies, Santa Clara, CA, USA) and pictures taken using an Andor Revolution XDi WD Spinning Disk Confocal Microscope (Oxford Instruments, Belfast, United Kingdom) equipped with an iXon Ultra 888 EMCCD Camera. Objective: 40×/1.3 UPlanFl N Oil, WD: 0.20 mm, ∞/0.17/FN26.5, UIS2. Laser lines used: 488 nm (50 mW) diode, 561 nm (50 mW) diode, 405 nm (100 mw) diode. Fluorescent filter cubes (excitation/emission) on a microscope (Olympus America, Center Valley, PA, USA): DAPI (350 nm/460 nm), FITC/GFP (480 nm/535 nm), TRITC/RFP/Cy3 (540 nm/605 nm). Five different fields of pictures were taken for each well with same intensities and laser power for all cell lines. Images were then processed and analysed for quantification of foci/cell with iMaris 9.2 Image Analysis Software.

### 4.7. Alkaline Comet Assay

Comet Assay was performed using a CometAssay Kit (Cat#4250-050-K, Bio-Techne, Minneapolis, MN, USA), according to manufacturer’s instructions. Briefly, cells were seeded into 6-well plates and treated next day with 0.1 µM of doxorubicin for 30 min. After media change, cells were cultured for 2 or 24 h and then combined with molten LMAgarose (Cat#4250-0505-02, Bio-Techne) at a ratio of 1:10 and immediately spread onto both wells of a CometSlide (Cat#4250-050-03, Bio-Techne). Upon lysis with Lysis Solution (Cat#4250-050-01, Bio-Techne) and immersion in Alkaline Unwinding Solution (200 mM NaOH, 1 mM EDTA, pH > 13), slides were then placed into the electrophoresis slide tray of the CometAssay ES Unit (Cat#4250-050-ES, Bio-Techne) covered with Alkaline Electrophoresis Solution (200 mM NaOH, 1 mM EDTA, pH > 13). After electrophoresis, slides were washed, dried, stained with SYBR Gold (Cat# S11494, Thermo Fisher Scientific, Waltham, MA, USA) and scanned using 10× objective with the Cytation 3 Cell Imaging Multi-Mode Reader (Cat#CYT3MV, Agilent). Each well was scanned using a 10 × 10 grid (100 images total), images processed and analysed using CometScore 2.0 to calculate % of DNA in tail and Tail length (µm).

### 4.8. Statistical Analysis

Statistical analyses were performed (with Prism 7, GraphPad, San Diego, CA, USA) using the unpaired Student’s *t* test with Welch’s correction and two-way ANOVA with Sidak’s post hoc test, as appropriate. For all experiments with error bars, S.D. was calculated to indicate the variation within each experiment and data, and values represent mean ± S.D.

## 5. Conclusions

In present study, we show that *Arid1a* is an essential survival gene whose loss impairs cellular growth, and can induce cell death. Thus, when lost during early stages of pancreatic tumorigenesis in mutant Ras-driven mouse models, it leads to cellular growth constraint, resulting in indolent low-grade cystic precursor lesions, such as IPMNs. However, with occurrence of an ‘escaper’ event(s), such as loss of p53, or upregulation of oncogenic networks like Myc and pluripotency factors, a subset of Arid1a-depleted precursor cells progress to adenocarcinomas. This provides novel opportunities to target such tumor cells in the clinic.

## Figures and Tables

**Figure 1 cancers-12-02695-f001:**
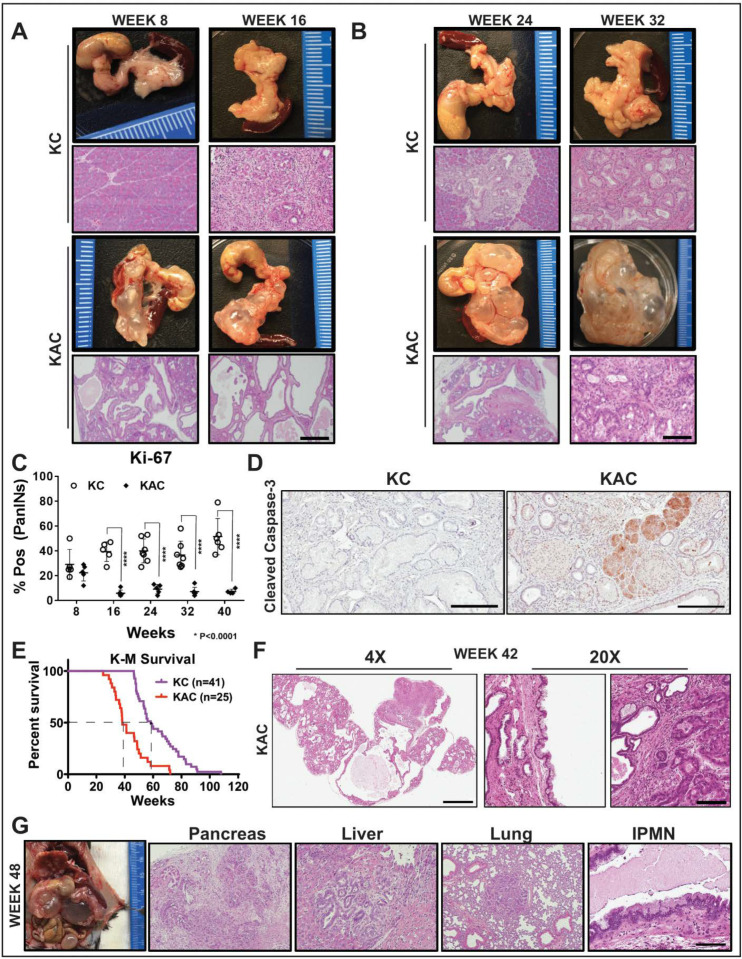
Conditional loss of Arid1a in mouse pancreas restricts growth of Kras-driven early premalignant lesions. (**A**,**B**) Representative gross macroscopic images and H&E stained histological sections from “KC” and “KAC” pancreata at early ages showing normal parenchyma replaced by mucinous cysts resembling human IPMNs. Representative images from at least 7 mice per age group are shown. (**C**) Quantification of immunohistochemical (IHC) staining for Ki-67 on pancreata from age-matched mice revealed lower percentage of proliferating PanINs lesions in “KAC” than “KC” group. For Ki67 quantification, % of Pos = (number of cells positive for Ki67 staining in PanINs)/(total number of cells in PanINs) from at least 5 mice per age group. (**D**) IHC staining for cleaved caspase-3 on 28-week old pancreata from “KC” and “KAC” mice showed stronger and widespread staining in acini and precursor lesions of “KAC” mice. (**E**) Kaplan-Meier survival curve for “KC” and “KAC” mice showing overall shorter median survival of “KAC” mice (38 weeks versus 58 weeks for “KC”) with *p* value of < 0.0001 based on Log-rank (Mantel-Cox) test. (**F**) Representative histopathological section from “KAC” mice characterized by foci of well-differentiated PDAC surrounded by areas of LG-IPMNs, ADMs and LG-PanINs (arrows). (**G**) Representative gross necropsy image and H&E stained sections from 48-week “KAC” mice showing well-differentiated PDAC (Pancreas), which metastasized to liver and lung, although majority of pancreas was still populated by IPMN lesions (IPMN). Scale bar is 100 µm.

**Figure 2 cancers-12-02695-f002:**
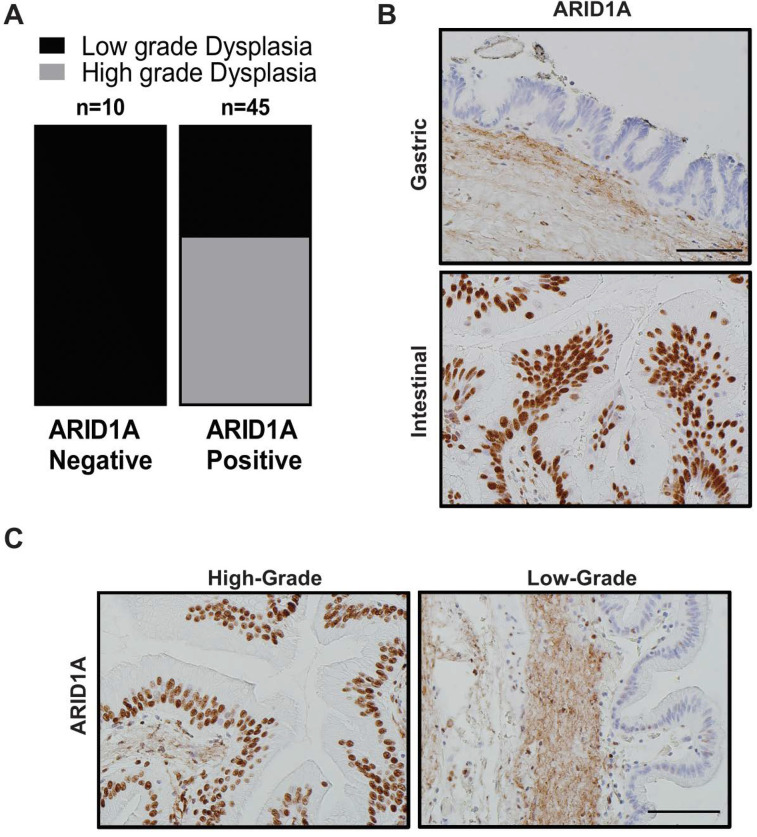
Loss of ARID1A expression among human IPMN is mostly restricted to low-grade gastric subtype IPMN. (**A**) Quantification of IHC staining for ARID1A on patient samples representing IPMNs of various subtypes and dysplasia grade revealed lost expression specifically in low-grade lesions of gastric subtype. (**B**) IHC staining for ARID1A in gastric versus intestinal subtype of human IPMN and in high-grade versus low-grade dysplasia in the same patient (**C**). Scale bar is 100 µm.

**Figure 3 cancers-12-02695-f003:**
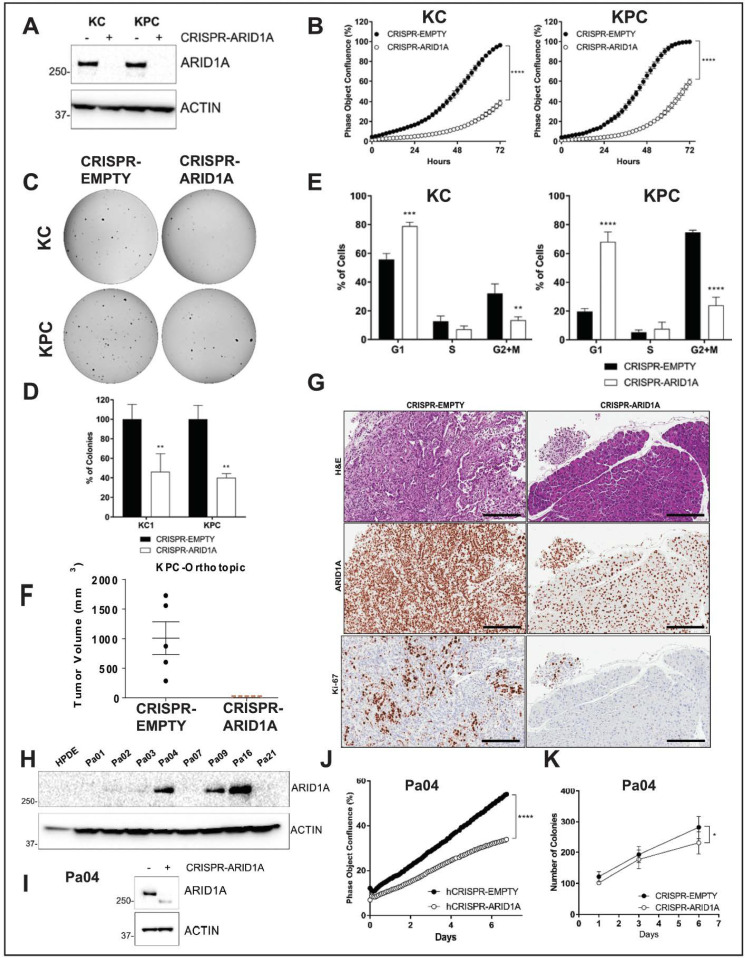
Arid1a loss is deleterious to growth in established PDAC cell lines. (**A**) Immunoblotting for ARID1A showed complete loss of expression in “KC” and “KPC” cell lines transfected with CRISPR/Cas9 targeting mouse *Arid1a*. (**B**) Monolayer culture in vitro showed reduced proliferation in *Arid1a*-deleted “KC” and “KPC” isogenic cell lines, represented as measure of culture confluence. Images were captured every 2 h using the live-imaging system (Incucyte ZOOM) and data represented as mean ± SD. ****, *p* < 0.0001 as determined by two-way ANOVA test. (**C**,**D**) Anchorage-independent colony growth assay on soft-agar showed reduced number of colonies in *Arid1a*-deleted KC and KPC isogenic cell lines after 2-weeks. Representative pictures of colonies stained with crystal violet are shown and the bar graph shows the % of colonies relative to the CRISPR-EMPTY control for each cell line. (**E**) Flow cytometric cell cycle analysis of PI-stained isogenic “KC” and “KPC” cell lines. Histogram indicates the mean of percentage of cells in each phase of the cell cycle from three independent experiments. (**F**) Orthotopically implanted isogenic “KPC” cells showed lack of growth in *Arid1a*-deleted cohort 50d post-implantation. Plot shows mean ± SD of primary tumor volume measured with digital caliper. (**G**) Representative images of H&E and IHC staining for ARID1A and Ki67 expression in sections from orthotopic tumors showed absence of growth of *Arid1a*-deleted cells. Scale bar is 100 µm. (**H**) Immunoblotting for ARID1A in a panel of patient-derived human cell lines confirming loss of expression by CRISPR-ARID1A in a cell line with endogenous expression (**I**). (**J**) Monolayer culture in vitro showed reduced proliferation in *Arid1a*-deleted Pa04 cells, represented as measure of culture confluence. Images were captured every 2 h using the live-imaging system (Incucyte ZOOM) and data represented as mean ± SD. ****, *p* < 0.0001 as determined by two-way ANOVA test. (**K**) Colony growth assay in soft-agar showed reduced number of colonies in *Arid1a*-deleted Pa04 cells, expressed as mean of total colonies/well. Representative findings from at least three independent experiments and data analyzed using the two-tailed unpaired Student’s *t* test and considered significant if *, *p* < 0.05; **, *p* < 0.01; ***, *p* < 0.001; ****, *p* < 0.0001, unless otherwise specified. Detailed information can be found at Appendix A.

**Figure 4 cancers-12-02695-f004:**
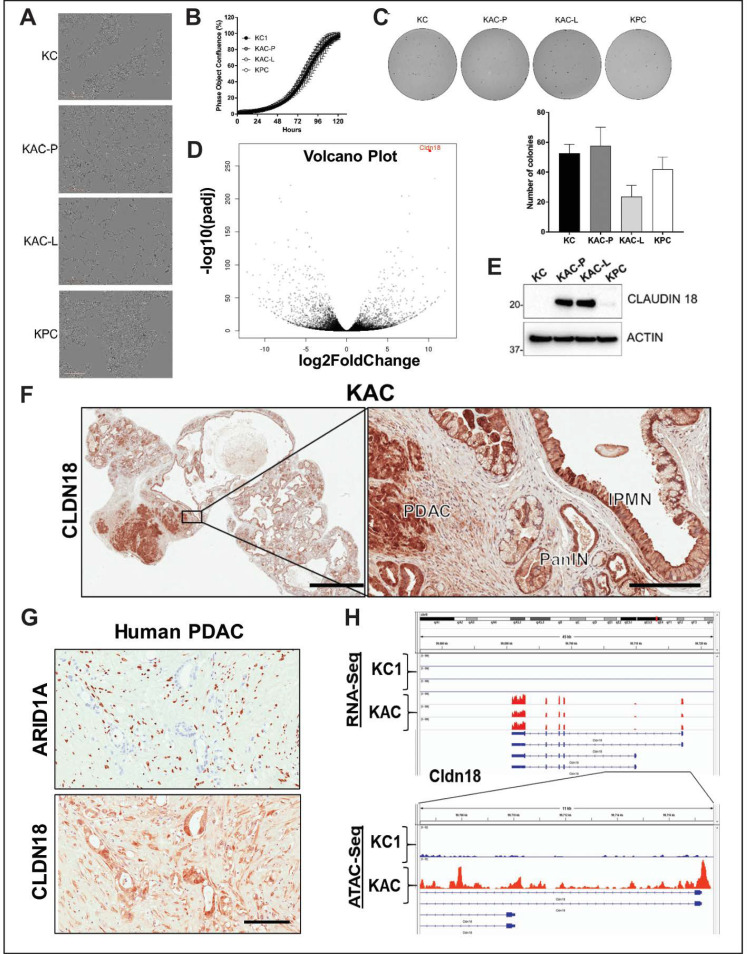
Characterization of autochthonous *Arid1a*-null PDAC cells from “KAC” mice. (**A**) In vitro monolayer cultures of “KC”, “KAC-P”, “KAC-L” and “KPC” cells revealed mesenchymal-like elongated morphology of “KAC” cells. Scale bar, 50 μm. (**B**,**C**) Assessment of growth in both monolayer (**B**) as well as soft-agar (**C**) showed no significant difference between “KAC” cells when compared to “KC” and “KPC”. Representative images of crystal violet stained colonies from 3 independent experiments. (**D**) Volcano plot of differentially expressed genes in “KAC-P” and “KC” cells, using RNA-Seq, showed Claudin 18 (CLDN18) as one of the top hit (*n* = 3). (**E**) Immunoblotting for mouse CLDN18 confirmed high expression levels in “KAC” PDAC cells compared to “KC” and “KPC”. Detailed information can be found at Appendix A. (**F**) IHC on pancreatic sections from “KAC” mice showed strong expression in epithelium of IPMN, PanIN, and PDAC, correlative with lack of ARID1A expression. *Left*, low magnification (2× objective) view of pancreatic section, *Right*, high magnification view (20× objective, Scale bar is 100 µm). (**G**) IHC on FFPE sections from human PDAC showed inverse relation between expression of ARID1A and CLDN18. Scale bar is 100 µm. (**H**) High transcript levels of Cldn18 in RNA-Seq corresponded to open chromatin at 5’ promoter region of *Cldn18* gene by ATAC-Seq on “KC and “KAC-P” cells (*n* = 3).

**Figure 5 cancers-12-02695-f005:**
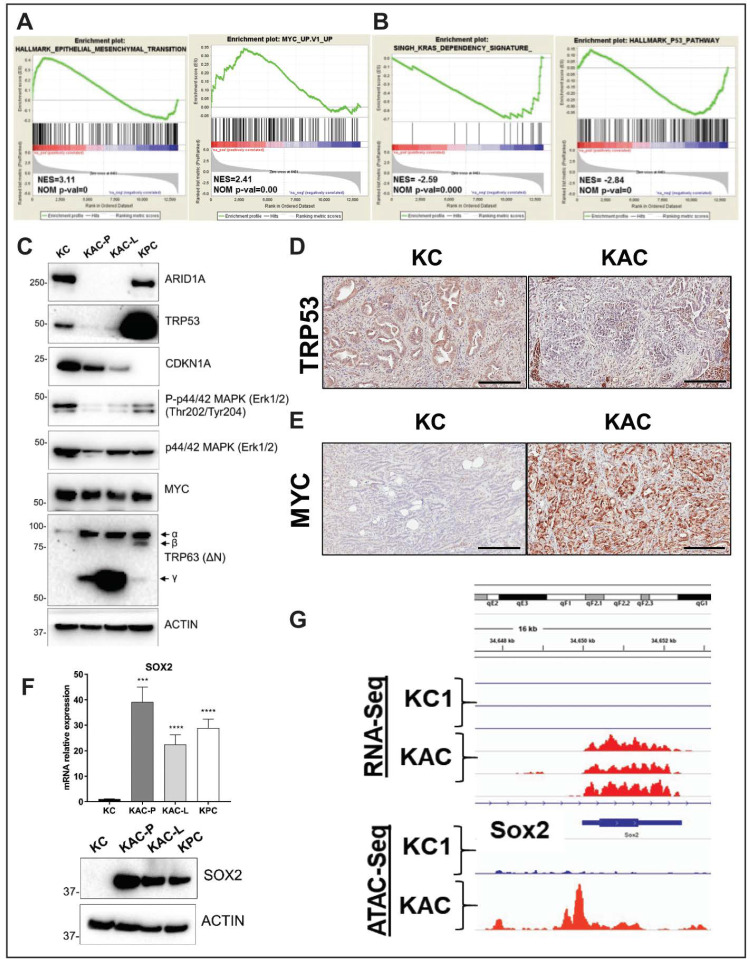
Identification of oncogenic mechanisms in “KAC” PDAC cells. (**A**,**B**) Representative plots from GSEA analysis of differentially expressed genes (2-fold change) identified from RNA-seq on “KC” and “KPC-P” cells revealed positive enrichment in hallmark of EMT and Myc target genes and negative enrichment in Kras dependency gene signature and hallmark P53 pathway. (**C**) Immunoblotting on protein lysates from autochthonous mouse PDAC cell lines showed loss of ARID1A and TRP53 expression in “KAC” cells along with reduction in levels of CDKN1A (p21) and phospho-Erk1/2. While total MYC levels were unchanged in all lines, “KAC” cells showed high levels of Trp63 (δN) isoform γ as compared to “KC” and “KPC” cells. (**D**–**E**) Representative microscopic images of IHC for TRP53 and MYC on PDAC sections from > 1-year old “KC” and “KAC” mice showed focal loss of p53 expression and strong expression of MYC in ARID1A-negative “KAC” group. Scale bar is 100 µm. (**F**) High expression of pluripotency-associated transcription factor SOX2 in “KAC” cell lines was validated by both RT^2^-PCR (top) and immunoblotting (bottom). Detailed information can be found at Appendix A. (**G**) High transcript levels of Sox2 in RNA-Seq corresponded to open chromatin at 5’ promoter region of *Sox2* gene by ATAC-Seq on “KC and “KAPC-P” cells (*n* = 3). ***, *p* < 0.001; ****, *p* < 0.0001.

**Figure 6 cancers-12-02695-f006:**
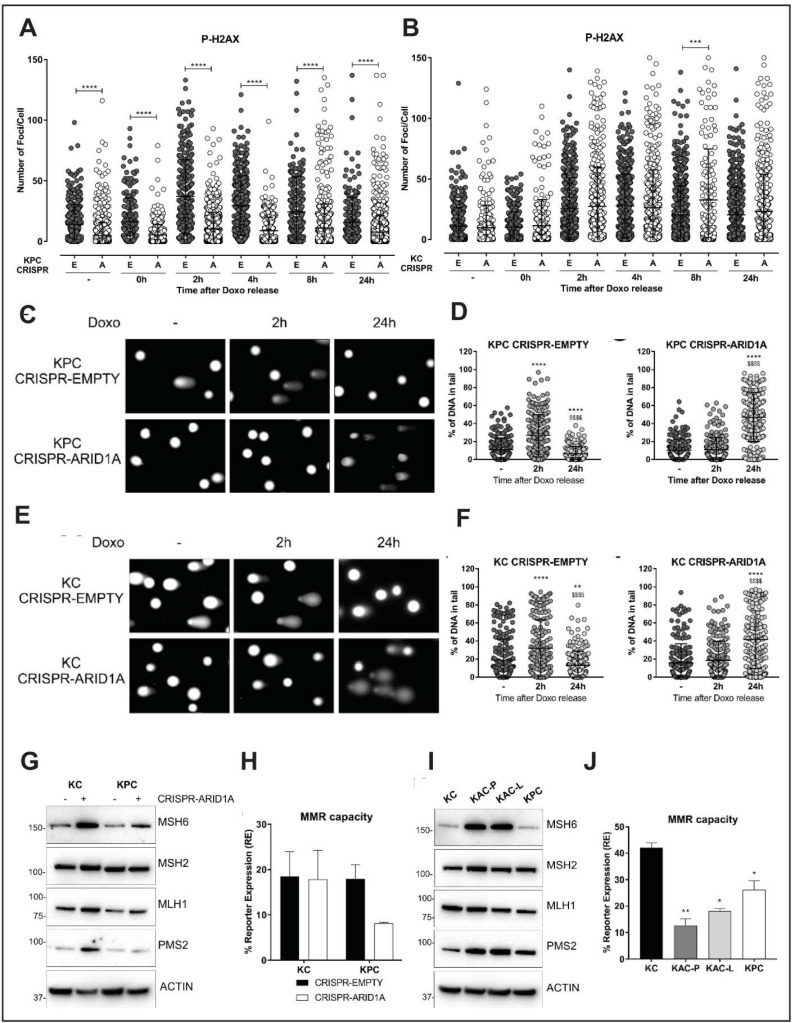
Loss of Arid1a impairs DNA damage and mismatch repair. (**A**,**B**) Quantification of immunofluorescent staining for phospho-gH2AX in isogenic pair of “KPC” (**A**) or “KC” (**B**) cell lines transfected with either empty vector (**E**) or Arid1a-targeting CRISPR (**A**), and exposed to DNA damaging agent like 0.1 µM Doxorubicin (Doxo) for 30 min, then released for the indicated time points. Graph representative of 3 independent experiments, show the quantification of number of foci/cell performed with iMaris Microscopy Image Analysis Software (Bitplane) and showing lower number of p-gH2AX positive foci only in *Arid1a*-deleted “KPC” cells but not “KC” cohort. (**C**–**F**) Comet assay using Doxo treated isogenic pairs of “KPC “(**C**,**D**) and “KC” (**E**,**F**) cells lines showed impaired DNA damage repair in *Arid1a*-deleted cohort of both pairs. Representative images (left panels) and scatter plots with quantification of the comet tail intensity relative to the head, expressed in % of DNA in tail (right panels) are shown. * two-tailed unpaired Student’s *t* test against untreated sample, ^$^ two-tailed unpaired Student’s *t* test against 2 h timepoint. (**G**,**I**)**,** Immunoblotting for MMR proteins on lysates from isogenic (**G**) or autochthonous (**I**) cell lines, with or without *Arid1a*-deletion, revealed higher expression of MSH6 and PMS2 in *Arid1a*-deleted cells. Detailed information can be found at Appendix A. (**H**,**J**) A fluorescence-based multiplex flow-cytometric host cell reactivation assay (FM-HCR) that measures the ability of cultured cells to repair plasmid reporters bearing mismatch, showed defective MMR in *Arid1a*-deleted cells, expressed as % of reporter expression. Three independent experiments were conducted and data represented as mean ± SD and two-tailed unpaired Student’s *t* test have been used for data analysis (unless otherwise indicated) and considered significant if *, *p* < 0.05; **, *p* < 0.01; ***, *p* < 0.001; ****, *p* < 0.0001; $$$$, *p* < 0.0001, unless otherwise specified.

**Figure 7 cancers-12-02695-f007:**
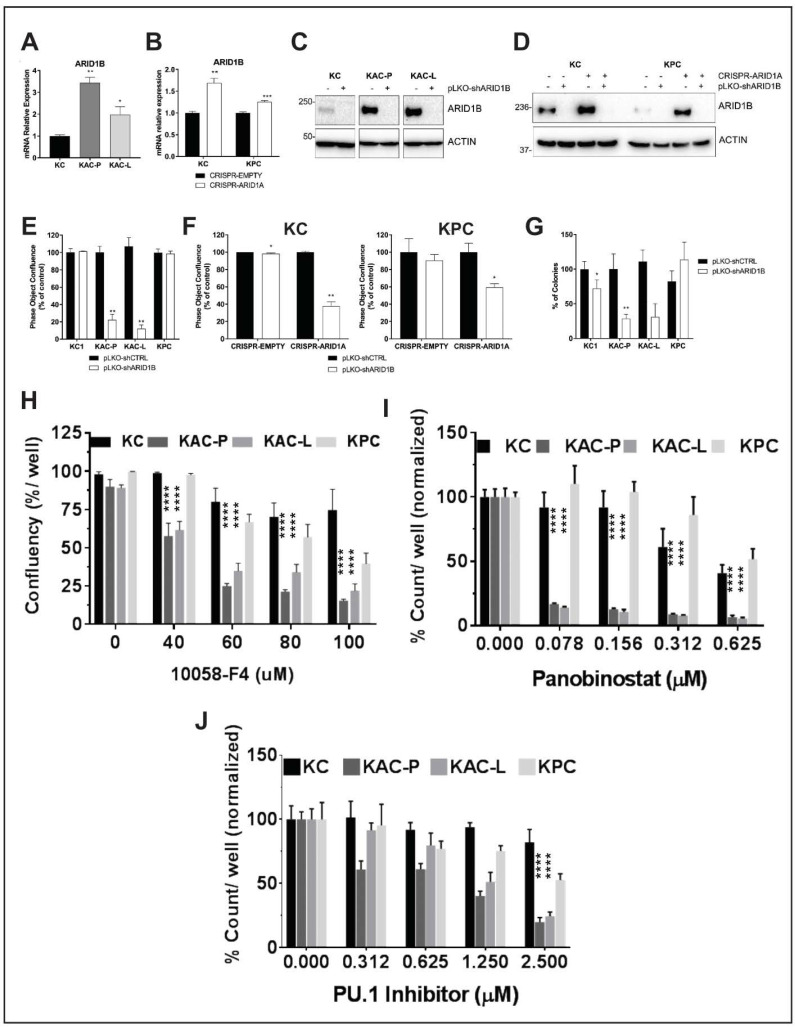
Identification of synthetic lethality in autochthonous *Arid1a*-null PDAC cells. (**A**,**B**) Semi-quantitative RT^2^ PCR revealed higher Arid1b mRNA expression in *Arid1a*-deleted autochthonous (**A**) and isogenic (**B**) cell lines, compared to “KC” or CRISPR-EMPTY control cells, respectively. (**C**,**D**) Immunoblotting for ARID1B confirmed loss of protein expression in *Arid1a*-deleted autochthonous (**C**) and isogenic (**D**) cell lines after Arid1b knockdown using pLKO-shRNA. Detailed information can be found at Appendix A. (**E**,**F**) In vitro monolayer cultures of autochthonous (**E**) and isogenic “KC” and “KPC” (**F**) cell lines upon knockdown of Arid1b, showed significant reduction in proliferation, expressed as % of pLKO-shCTRL-transduced cells. (**G**) Anchorage-independent colony growth assay on soft-agar showed significant reduction in number of colonies upon Arid1b-knockdown, predominantly in *Arid1a*-deleted “KAC” cell lines. Bar graph shows the % of colonies, normalized on the pLKO-shCTRL for each cell line. (**H**–**J**) In vitro monolayer cultures of autochthonous PDAC cell lines were treated with Myc-inhibitor 10058-F4 (**H**), Panobinostat (**I**) or Pu.1 Inhibitor (**J**) at indicated doses for 72 h and their growth measured either as culture confluence (**H**) or cell count (**I**,**J**), normalized to vehicle-treated control. Images were captured every 2 h using the live-imaging system (Incucyte ZOOM) and data plotted as mean ± SD. Two-tailed unpaired Student’s *t* test was used for data analysis (unless otherwise indicated) and considered significant if *p* < 0.05; **, *p* < 0.01; ***, *p* < 0.001; ****, *p* < 0.0001.

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
