# Peer review of "Paradoxical Role of AT-rich Interactive Domain 1A in Restraining Pancreatic Carcinogenesis"

_cancers, 2020, doi:10.3390/cancers12092695_

Round 1
Reviewer 1 Report
The identification of loss-of-function mutations in ARID1A in ~6% of PDAC cases suggests that it is a tumor suppressor gene that likely genetically cooperates with mutant KRAS in pancreatic tumorigenesis. Consistent with this hypothesis, established PDAC precursor lesions (i.e. intraductal papillary mucinous neoplasms (IPMNs)) were reported in a previously described mouse model containing pancreas-specific Arid1a loss together with mutant Kras expression. However, the hypothesis that ARID1A functions as a tumor suppressor gene is called into question by the fact that the progression of IPMNs to PDAC was only observed in 20% of these mice. Moreover, the rate of mutation in ARID1A or loss of ARI1A expression in human IPMN samples was much higher than in human PDAC samples. To further investigate the potential that ARID1A may act as a tumor suppressor gene in PDAC, the authors of this manuscript reassessed the effect of ARID1A loss in multistep pancreatic carcinogenesis in genetically engineered autochthonous mouse models of PDAC as well as established human PDAC cell lines. Based on their data, the authors’ propose that Arid1a is critical for the early progression of mutant Kras-driven pre-malignant lesions into PDAC. Specifically, they show reduced levels of proliferation and elevated levels of apoptosis in Ptf1a-Cre;KrasG12D;Arid1af/f (KAC) mice as compared to Ptf1a-Cre;KrasG12D (KC) mice. The authors also report that the CRISPR/Cas9-mediated deletion of ARID1A from established human PDAC cell lines causes the reduction of their growth and migration as well as the impairment of their ability to repair DNA damage. The early development of IPMNs was observed in the KC mice and a subset of those mice went on to develop aggressive PDAC with age. PDAC cells harvested from aged KAC mice revealed the presence of epigenetic alterations that the authors propose may underlie the various compensatory mechanisms used by PDAC cells to escape the growth impairment caused by the loss of Arid1a. Based on these findings, the authors conclude that Arid1a is an essential survival gene and that its expression is critical during the early stages of pancreatic tumorigenesis in mice. Overall, I feel that this is an intriguing manuscript that further reveals the complicated contribution of the loss of Arid1a function to pancreatic carcinogenesis. Unfortunately, there are several major and minor issues that significantly reduce my enthusiasm for this work. Consequently, I cannot recommend this manuscript for publication in its current state.
Major Issues:
- There is a general lack of proper controls throughout this work. Specifically, the results generated in the KC and KAC mice should also be compared with results generated in Arid1a+/+ and Arid1af/flittermate mice. It is also important that the authors test if the re-expression of wild type Arid1a can rescue the phenotypes reported in the CRISPR-Cas9-generated Arid1a-null or Arid1a-targetted shRNA construct-expressing PDAC cell lines. This second set of controls is critical for being able to rule-out the involvement of potential off-target effects caused by either the CRISPR-Cas9 or shRNA constructs in the PDAC cell lines. In addition, the authors should provide images of Arid1a staining in tissues from a non-PDAC patient for comparison in Figures 2B and 2C or control mice in Figures 4F and 4G.
- Many of the results of the experiments shown in this work were not quantitatively analyzed. As a result, it is difficult to judge the robustness of these results. Specific examples are identified below.
- Figure 1 panels A, B, D, F, and G
- Figure 3 panel C
- Figure 4 panels A, C, F, and G
- Figure 5 panels C, D, and E
- Figure 6 panels G and I
- The quality of the cell images presented in Figures 3C, 4A, 4C is low, which makes them rather uninterruptable.
- The presence of the smaller band shown in the CRISPR-ARID1A lane of Figure 3I is worrisome. Does this indicate that the CRISPR-ARID1A construct expressing Pa04 cells are not null for Arid1a, but express a truncated version of Arid1a? This information is critical for the proper interpretation of the results generated using this cell line as well as for future scientists to potentially reproduce the authors’ results.
Minor Issues:
- Gene naming issues: Human gene names are capitalized and italicized. The first letter of mouse gene names is capitalized and italicized. There are several inconsistencies throughout the manuscript that need to be addressed.
- More information regarding the Andor Revolution XDi WD Spinning Disk Confocal Microscope used in this work is needed. Specifically, I need to know the objective magnification, N.A., and level of correction as well as the laser lines, the excitation/emission filters, and the camera used to generate the data shown in this manuscript.
- What are the magnification, N.A., and level of correction of the objective used with the Cytation 3 Cell Imaging Multi-mode Reader?
- Figure 1F: Arrows should be used to show the LG-IPMNs, ADMs, and LG-PanINs in these images.
- Figure 2A: What exactly is being shown here? No units are displayed.
- It should be clearly stated in the figures or their legends the number of times or biological replicates were performed for a given experiment.
- The authors need to include scale bars in all images, or at least one image in a panel of images.
Author Response
Reviewer 1
We appreciate the detailed comments from Reviewer#1 and glad with his/her overall impression that this manuscript further reveals the complicated contribution of the loss of Arid1a function to pancreatic carcinogenesis.
Please see the point-wise response to all major and minor concerns raised (in bold):
Major Issues:
- There is a general lack of proper controls throughout this work. Specifically, the results generated in the KC and KAC mice should also be compared with results generated in Arid1a+/+ and Arid1af/flittermate mice.
Although, we have used all the appropriate controls in this study, some of them are not shown in figures for the sake of easy presentation of data. For example, Arid1a+/+ (ARID1a+/+;Ptf1a-Cre) and Arid1af/f (ARID1afl/fl;Ptf1a-Cre or “AC”) littermate mice were always observed as control alongside experimental cohorts (“KC” and “KAC”). While observations from “AC” mice are presented in the manuscript (Page 2, lines 72-77), Arid1a+/+ mice were found to be no different than the wild type B6 mice, as expected from all previously published literature using Ptf1a-Cre mice. Importantly, in order to evaluate functional role of Arid1a mutation in experimental cohort of “KAC” mice, we used “KC” mice as control, since both of these cohorts have mutant Kras expression as common factor, differing only in expression of Arid1a . Additionally, we maintain that observations from these control mice are not different from what was reported in other parallel but independent published manuscripts.
It is also important that the authors test if the re-expression of wild type Arid1a can rescue the phenotypes reported in the CRISPR-Cas9-generated Arid1a-null or Arid1a-targetted shRNA construct-expressing PDAC cell lines.
We agree that such rescue experiments would have been useful; however, there are significant technical challenges involved. We repeatedly observed that loss of Arid1a expression in both murine and human PDAC cell lines led to reduced cellular proliferation and increased cell death (Figure 3), thus not allowing any additional experiments involving long-term culture, such as stable re-expression of wild type Arid1a and selection of positive single-cell clones. Indeed, when we attempted to establish long-term culture of these cells, it was only enabled by growth of cells that escaped CRISPR-cas9 mediated gene deletion, as evident by detection of Arid1a expression by Western Blotting in long-term cultures.
This second set of controls is critical for being able to rule-out the involvement of potential off-target effects caused by either the CRISPR-Cas9 or shRNA constructs in the PDAC cell lines. In addition, the authors should provide images of Arid1a staining in tissues from a non-PDAC patient for comparison in Figures 2B and 2C or control mice in Figures 4F and 4G.
We understand the reviewer suggested that we add images of Arid1a staining in tissues from a non-PDAC patient for comparison in Figures 2. The emphasis of Figure 2 is to show that loss of ARID1A expression is mostly restricted to low-grade IPMN lesions of gastric subtype as compared to high-grade lesions. Even in the same patient, this loss of ARID1A expression was restricted to low-grade dysplasia. Thus, the appropriate control in these figure panels are lesions of IPMNs rather than from non-PDAC patient. We did IHC staining for ARID1A on variety of human tissues and found similar expression among them, including pancreas. These images were not included in the manuscript, since normal ARID1A expression is well reported in published literature.
It is not clear if the reviewer is asking about ARID1A staining in control mice for Figures 4F and 4G. Figure 4F show CLDN18 staining in pancreatic sections from “KAC” mice to emphasize the strong expression of CLDN18 in corresponding Arid1a-null cells from “KAC” mice. Supplemental Fig S2B already shows lack of Arid1a expression in pancreatic epithelial cells from “KAC” mice. If reviewer is interested in CLDN18 staining in control mice, we are providing an additional file showing staining for CLDN18 in pancreatic sections from age-matched control “KC” mice.
Figures 4G shows correlation between ARID1A and CLDN18 staining on human PDAC sections, so there are no control mice involved.
Many of the results of the experiments shown in this work were not quantitatively analyzed. As a result, it is difficult to judge the robustness of these results. Specific examples are identified below.
- Figure 1 panels A, B, D, F, and G
- Figure 3 panel C
- Figure 4 panels A, C, F, and G
- Figure 5 panels C, D, and E
- Figure 6 panels G and I
Here are the pointwise reply regarding quantification of data:
- Figure 1 panels A, B, D, F, and G: These panels are meant for qualitative analysis, since they show marked histological changes in the pancreatic architecture, as assessed by a board-certified pathologist.
- Figure 3 panel C: Quantification is shown in panel D.
- Figure 4 panels A, C, F, and G: Panel A is representative images of cellular morphology of various autochthonous mouse cell lines, so no quantification is needed. Quanitification of Panel C is shown as bar graph below the colony growth images. Panel F shows strong CLDN18 staining on Arid1a-null epithelial cells from “KAC” mice, while Panel G shows inverse correlation between ARID1A and CLDN18 staining on human PDAC sections.
- Figure 5 panels C, D, and E: We had submitted a separate supplementary file providing densitometry analysis on all western blots. Panel D and E shows clear qualitative differences in expression of TRP53 and MYC between pancreatic sections from “KAC” and control “KC” mice.
- Figure 6 panels G and I: We had submitted a separate supplementary file providing densitometry analysis on all western blots.
In short, for all figure panels showing Western Blot data, we had submitted a separate supplementary file providing densitometry analysis and change in relative expression (as compared to control sample). For all histological images (H&E and IHC) showing qualitative data, we presented representative images from formalin-fixed pancreas from at least 7 mice per age group. The FFPE tissue from each mouse was sectioned at various depths (100um apart), each sections was stained and multiple images were captured from each sections. We have also mentioned the number of mice used to obtain representative images, for each age group, in the Result section 2.1.
- The quality of the cell images presented in Figures 3C, 4A, 4C is low, which makes them rather uninterruptable.
Quantification of Figure 3C and 4C are also shown in the figures. By increasing the magnification of submitted mansucript, we can see these panels better without affecting pixel resolution. However, we can provide separate high resolution images if required.
- The presence of the smaller band shown in the CRISPR-ARID1A lane of Figure 3I is worrisome. Does this indicate that the CRISPR-ARID1A construct expressing Pa04 cells are not null for Arid1a, but express a truncated version of Arid1a? This information is critical for the proper interpretation of the results generated using this cell line as well as for future scientists to potentially reproduce the authors’ results.
The smaller band seen in the CRISPR-ARID1A lane of Figure 3I is a background signal anomaly, seen inconsistently in even control samples across various independent western blots. For clarity, we have provided a substituted panel from an independent experiment and updated the Figure 3.
Minor Issues:
- Gene naming issues: Human gene names are capitalized and italicized. The first letter of mouse gene names is capitalized and italicized. There are several inconsistencies throughout the manuscript that need to be addressed.
Human genes were listed ABC while mouse genes were listed as Abc.
- More information regarding the Andor Revolution XDi WD Spinning Disk Confocal Microscope used in this work is needed. Specifically, I need to know the objective magnification, N.A., and level of correction as well as the laser lines, the excitation/emission filters, and the camera used to generate the data shown in this manuscript.
We are providing the requested details as follow: iXon Ultra 888 EMCCD Camera. Objective: 40x/1.3 UPlanFl N Oil, WD: 0.20mm, ∞/0.17/FN26.5, UIS2. Laser lines used: 488 nm (50 mW) diode, 561 nm (50 mW) diode, 405 nm (100 mw) diode. Fluorescent filter cubes (excitation/emission) on Olympus microscope: DAPI (350 nm/460 nm), FITC/GFP (480 nm/535 nm), TRITC/RFP/Cy3 (540 nm/605nm)
- What are the magnification, N.A., and level of correction of the objective used with the Cytation 3 Cell Imaging Multi-mode Reader?
4x, N.A. 0.1, ACH
- Figure 1F: Arrows should be used to show the LG-IPMNs, ADMs, and LG-PanINs in these images.
We have added the arrows in Figure 1F, and updated the figure legend accordingly.
- Figure 2A: What exactly is being shown here? No units are displayed.
It is a ‘part of the whole’ section chart showing distribution of IPMN cases with low-grade and high-grade dysplasia when separated based on ARID1A staining. Among ARID1A-negative cases, all 10 cases showed low-grade dysaplasia, while, among ARID1A-positive cases, 18/45 showed low-grade and 27/45 showed high-grade dysplasia.
- It should be clearly stated in the figures or their legends the number of times or biological replicates were performed for a given experiment.
We have provided that information. For example, figure 3: “Representative findings from at least 3 independent experiments and data analyzed using the two-tailed unpaired Student’s t test and considered significant if *, P< 0.05; **, P< 0.01; ***, P< 0.001; ****, P< 0.0001, unless otherwise specified.”
- The authors need to include scale bars in all images, or at least one image in a panel of images.
We checked and made sure all Figure Panels that required a scale bar, have it and scale is explained in the figure legend.

Reviewer 2 Report
Peer-review of “Paradoxical role of AT-rich interactive domain 1A in restraining pancreatic carcinogenesis
Several studies to decipher the role of ARID1A in PDAC have recently been published. However, some of these studies present paradoxical results. Thus, there is a need to reconcile the functions of ARID1A. Therefore, the aim of this study was to clarify the functional role of ARID1A in PDAC by analyzing various genetically engineered PDAC mouse models and human and murine cell lines. Briefly, the authors showed that ARID1A is critical during early stages of pancreatic tumorigenesis in mouse models and that deletion of ARID1a in established PDAC cell lines reduced cell growth and migration and compromised DNA damage repair.
Questions:
- Despite the need for clarification on the role of ARID1A, a major concern of the findings in this manuscript is the lack of novelty.
- Indeed, in the paper from Kimura et al. (authors’ reference #3) it has been previously shown that loss of ARID1A downregulates the mTOR pathway, one of the downstream effects being the reduction of proliferation and cell survival. What is the novel finding when the authors show that proliferation and cell survival is impaired by ARID1A loss?
- Moreover, in the paper from Wang et al. (authors’ reference #5) it has already been shown that loss of ARID1A induced development of mucinous cysts and reduced the survival of mice. Was it necessary to show this again?
- In order to assess the full loss of ARID1A expression in KAC mice, the authors’ performed IHC. They state that “Lack of Arid1a expression in the precursor lesions arising in the “KAC” mice was confirmed by IHC”. They should specify that ARID1A expression was absent exclusively in epithelial cells.
- Concerning the abstract, its main focus is on the experiments conducted in mice and their resulting cell lines. However, the studies also included experiments using human cell lines, these results should also be mentioned in the abstract. In addition, the “escaper” mechanisms are key findings of this study and should also be included in the abstract.
- In line 41, the authors state that the “escaper” mechanisms are of epigenetic origin. In the result section 2.4, they identified p53 and MYC expression being impacted by loss of ARID1A. However, are these changes in expression really of epigenetic origin and not additional mutations? Could these be due to Kras itself? This could be clarified in additional experiments.
- The authors observed that in mice of growing age, the effect of ARID1A deficiency is lost or gets bypassed; is the effect of ARID1A age-dependent?
- As the experimental systems used and results generated in this study are complex and to a certain degree paradoxical, it is easy to get lost. We suggest to include a schematic overview of the key findings of this study, this could be very helpful.
Minor observations
- In line 27, there should be an “and” instead of “,” between “GEM model” and “PDAC cell lines”. Moreover, a space is missing before “Methods”.
- In line 31, “cells lines” should be replace by “cell lines”.
- In line 43, a space is missing before “Conclusions”.
- In section 2.3, line 150 “KPC” mice should be briefly defined here
- In line 202, the “J” referring to figure J should be bold.
- In section 2.4 of the results, they identify Cldn18 as differentially expressed between “KAC-P” and “KC” cells. Cldn18 should be briefly introduced here.
Author Response
Reviewer#2
Several studies to decipher the role of ARID1A in PDAC have recently been published. However, some of these studies present paradoxical results. Thus, there is a need to reconcile the functions of ARID1A. Therefore, the aim of this study was to clarify the functional role of ARID1A in PDAC by analyzing various genetically engineered PDAC mouse models and human and murine cell lines. Briefly, the authors showed that ARID1A is critical during early stages of pancreatic tumorigenesis in mouse models and that deletion of ARID1a in established PDAC cell lines reduced cell growth and migration and compromised DNA damage repair.
We appreciate the detailed comments from Reviewer#2 and glad with his/her overall impression that there is an urgent need to reconcile published functions of ARID1A due to paradoxical results and that this manuscript further reveals the complicated contribution of the loss of Arid1a function to pancreatic carcinogenesis.
Please see the point-wise response to all major and minor concerns raised (in bold):
Questions:
- Despite the need for clarification on the role of ARID1A, a major concern of the findings in this manuscript is the lack of novelty.
- Indeed, in the paper from Kimura et al. (authors’ reference #3) it has been previously shown that loss of ARID1A downregulates the mTOR pathway, one of the downstream effects being the reduction of proliferation and cell survival. What is the novel finding when the authors show that proliferation and cell survival is impaired by ARID1A loss?
While Kimura et. al (Ref#3) also reported few findings similar to our manuscript such as lower proliferation index (Ki-67 staining) in cystic neoplasms and PanINs of KAC mice than PanINs of KC mice. However, they characterized their IPMNs to be of pancreatobiliary- or oncocytic subtype, in contrast to gastric subtype noted by us and other groups (Ref#5 and 6). It is noteworthy that pancreatobiliary subtype IPMN show higher grade of dyspasia than gastric subtype. Moreover, Kimura et al. despite reporting that subset of KAC mice showed invasive PDAC after 48 weeks (in agreement with our data), they did not show any possible escaper mechanisms.
- Moreover, in the paper from Wang et al. (authors’ reference #5) it has already been shown that loss of ARID1A induced development of mucinous cysts and reduced the survival of mice. Was it necessary to show this again?
While, Wang et al (Ref#5) reported similar kinds of IPMN lesions as us, there have been some discrepancies in subtype of IPMNs reported (high-grade versus low-grade) by other groups (such as Ref#3). We also provided additional data that loss of ARID1A expression was restricted to low-grade dysplasia, supporting our GEM model data that loss of Arid1a constrain the progression of low grade IPMNs into higher-grade lesions.
- In order to assess the full loss of ARID1A expression in KAC mice, the authors’ performed IHC. They state that “Lack of Arid1a expression in the precursor lesions arising in the “KAC” mice was confirmed by IHC”. They should specify that ARID1A expression was absent exclusively in epithelial cells.
We have now specified in the main text that loss of ARID1A expression in “KAC” mice was restricted to epithelial cells (precursor lesions). (Page 3, Lines 101-02)
- Concerning the abstract, its main focus is on the experiments conducted in mice and their resulting cell lines. However, the studies also included experiments using human cell lines, these results should also be mentioned in the abstract. In addition, the “escaper” mechanisms are key findings of this study and should also be included in the abstract.
We have now revised the abstract accordingly.
- In line 41, the authors state that the “escaper” mechanisms are of epigenetic origin. In the result section 2.4, they identified p53 and MYC expression being impacted by loss of ARID1A. However, are these changes in expression really of epigenetic origin and not additional mutations? Could these be due to Kras itself? This could be clarified in additional experiments.
Since, we did not sequence the genome of tumors from “KAC” and “KC” mice, we cannot completely rule out mutations in Trp53 and Myc. We believe this scenario is unlikely, since we did not find any significant correlation between Arid1a mutations and MYC amplification or deleterious Trp53 mutation in human TCGA and COSMIC databases. Additionally, deleterious mutation in Trp53 gene in “KAC” would likely lead to loss of transcript expression. However, we did not observe any significant change in the transcript levels of Trp53 (quantified by RT2-PCR) between “KC” and “KAC” cells (Fig S4E). We did however, observed a substantial increase in levels of TRP63 RNA and protein expression of ΔNp63α and γ isoforms in “KAC” cells compared to “KC” and “KPC” cells (Figure 5C, Supplementary Fig. S4E), which has been shown to positively regulate MYC function [11]. MYC transcription has also been reported to be repressed by p53 and the loss of p53 synergistically enhances the Myc–induced tumorigenesis[38]. This would suggest that MYC function, at least in part, is regulated at epigenetic level. We also do not think these ‘escaper’ mechanisms are due to mutant Kras itself, since these were missing in “KC” cohort. MYC can also substitute as a pivotal driver in the setting of “Ras independence”, and the resulting tumors tend to be highly aggressive and chemoresistant [42,43].
- The authors observed that in mice of growing age, the effect of ARID1A deficiency is lost or gets bypassed; is the effect of ARID1A age-dependent?
We do not think effect of Arid1a loss is age-dependent but rather the result of tumor evolution (in case of mice with additional mutant Kras expression), which results in accumulation of various ‘escaper’ mechanisms which help bypass the growth restrain due to loss of Arid1a expression. In case of mice with just loss of ARID1A (“AC”), we see a progressive loss of pancreatic exocrine acinar cells with age, since in post-natal mice, expression of Cre (driven by Ptf1a promoter) is mostly restricted to acinar cells.
- As the experimental systems used and results generated in this study are complex and to a certain degree paradoxical, it is easy to get lost. We suggest to include a schematic overview of the key findings of this study, this could be very helpful.
We have now provided a graphical abstract to summarize the overview of key findings of this study.
Minor observations
- In line 27, there should be an “and” instead of “,” between “GEM model” and “PDAC cell lines”. Moreover, a space is missing before “Methods”.
- In line 31, “cells lines” should be replace by “cell lines”.
- In line 43, a space is missing before “Conclusions”.
- In section 2.3, line 150 “KPC” mice should be briefly defined here
- In line 202, the “J” referring to figure J should be bold.
- In section 2.4 of the results, they identify Cldn18 as differentially expressed between “KAC-P” and “KC” cells. Cldn18 should be briefly introduced here.
We have now taken care of these minor concerns in the revised manuscript.

Round 2
Reviewer 1 Report
I appreciate the authors attempt to address the issues that I raised in my review of their original manuscript. Overall, I feel that they have addressed the majority of my previously indicated issues. However, I still have the following concerns.
- I appreciate the technical challenges involved in my proposed rescue experiments for the shRNA and CRISPR-Cas9 experiments. However, it remains to be shown that the phenotypes reported are not due to unanticipated off-target effects. In the absence of rescue experiments, another way to control for the off-target effects of shRNA and CRISPR-Cas9 is to show that the same phenotype can be reproduced in cell lines generated using an independent shRNA construct or sgRNA.
- I would like the authors to increase the quality of the images in Figure 4A. Without these images, I cannot assess the “elongated morphology” of the KAC cells described by the authors.
- I would like the authors to include the information that they provided regarding their imaging systems in the Materials and Methods section of their manuscript.
I would like to see these concerns dealt with before I can accept this manuscript for publication.
